# Gaussian Mutual Information Maximization for Efficient Graph Self-Supervised Learning: Bridging Contrastive-based to Decorrelation-based

## ABSTRACT

Enlightened by the *InfoMax* principle, Graph Contrastive Learning (GCL) has achieved remarkable performance in processing large amounts of unlabeled graph data. Due to the impracticality of precisely calculating mutual information (MI), conventional contrastive methods turn to approximate its lower bound using parametric neural estimators, which inevitably introduces additional parameters and leads to increased computational complexity. Building upon a common Gaussian assumption on the distribution of node representations, a computationally tractable surrogate for the original MI can be rigorously derived, termed as Gaussian Mutual Information (GMI). Leveraging multi-view priors of GCL, we induce an efficient contrastive objective based on GMI with performance guarantees, eliminating the reliance on parameterized estimators and negative samples. The emergence of another decorrelation-based self-supervised learning branch parallels contrastive-based approaches. By positioning the proposed GMI-based objective as a pivot, we bridge the gap between these two research areas from two aspects of approximate form and consistent solution, which contributes to the advancement of a unified theoretical framework for self-supervised learning. Extensive comparison experiments, ablation studies, and visual analysis provide compelling evidence for the effectiveness and efficiency of our method while supporting our theoretical achievements.

## CCS CONCEPTS

• **Computing methodologies** → *Unsupervised learning*.

## KEYWORDS

Gaussian Mutual Information Maximization, Graph Self-Supervised Learning, Dimensional Collapse, Unified Theoretical Framework

## 1 INTRODUCTION

The scarcity of task-related annotations for graph data, which usually rely on domain knowledge and specific equipment such as chemical instruments [29], urgently calls for the emergence of advanced unsupervised learning methods without manual supervision. In this context, graph self-supervised learning (SSL) [6, 46, 55,

*ACM MM, 2024, Melbourne, Australia*
© 2024 Copyright held by the owner/author(s). Publication rights licensed to ACM.
ACM ISBN 978-x-xxxx-xxxx-x/YY/MM
https://doi.org/10.1145/nnnnnnn.nnnnnnn

58, 59] arises naturally in response to the prevailing demand, approaching and even surpassing the performance of their supervised counterparts [19, 24, 48]. SSL models are trained on well-designed pretext objectives in a task-agnostic manner, whose optimization results in general, meaningful, and transferable representations for downstream applications. As a distinguished member of the SSL family, multi-view learning with Siamese networks [27] has demonstrated exceptional performance and garnered widespread interest. At the heart of such methods is to extract invariant or common information from various augmented views of the same instance (*i.e.*, positive pairs) while adopting specific strategies to prevent model collapse. The existing multi-view learning methods can be classified into two distinct categories based on their means of addressing degradation: *contrastive* [20, 55, 58, 59] and *non-contrastive* [6, 46, 57] approaches. The former suppresses encoded representations from collapsing into a constant point by pushing negative pairs apart, while the latter employs special strategies such as decorrelating different representation dimensions [6, 57] or designing asymmetric network architecture [17, 46].

The concept of *contrastive* multi-view learning originates from information theory [2, 51], aiming to improve the consistency between various views by maximizing their mutual information (MI). Nevertheless, the exact computation of mutual information for high-dimensional continuous variables is usually intractable. To cope with this challenge, some previous endeavors have attempted to employ parameterized neural estimators to perform an empirical evaluation of mutual information from finite samples, yielding notable achievements like MINE [4], Jensen-Shannon estimator [31], and *InfoNCE* [18]. Formally, contrastive learning methods equipped with the parameterized estimator of MI manifest as a contrastiveness between positive pairs from a joint distribution and negative pairs from two marginal ones. Despite their decent performance, these methods are accompanied by several inherent drawbacks: a) a substantial number of samples are required to obtain reliable estimation and achieve satisfactory results, which inevitably increases computational burden; b) the incorporation of parameterized MI estimators amplifies the complexity of SSL models.

Deviating from the conventional graph contrastive learning methods, we delve into lightweight and efficient alternatives with no reliance on parameterized MI estimators for node-level representation learning. Assuming node representations obey a Gaussian distribution, a feasible closed-form solution can be obtained, called Gaussian Mutual Information (GMI) [36, 38], through tractable integration operations on the native definition of MI. In its mathematical form, the estimation of GMI exclusively depends on the covariance matrices, which can be effortlessly obtained from empirical data (*i.e.*, node representations). Independent of additional architectures, the resultant SSL objective under GMI can be directly calculated within the representation space, leading to higher

computational efficiency and better resource friendliness. Most importantly, the performance of the proposed method can still hold even when actual scenarios deviate from Gaussian distributions, thereby extending its applicability beyond Gaussian constraints.

As another indispensable branch of the SSL family, the decorrelation-based non-contrastive methods [3, 6, 12, 57] prevent degenerate solutions and learn diverse representations by decoupling various channels, whose objective functions exhibit an utterly distinct appearance from those of contrastive-based ones. While the distinctions between the two branches have been thoroughly discussed, their latent theoretical relationships remain enshrouded in ambiguity. Imposing a cross-view identity constraint, which enhances the perfect alignment of representations from different views of the same instance, to our proposed GMI-based objective function, we employ the newly induced objective as a pivot to elucidate the underlying connections between decorrelation-based and contrastive-based methods. On the one hand, the former is formally equivalent to a second-order Taylor series expansion of the latter. On the other hand, their objectives share consistent solutions. Overall, the decorrelation-based methods can be regarded as an instantiation of contrastive learning under the Gaussian assumption on the distribution of node representations and identity constraint.

Our contributions in this paper are summarized as follows:

- In light of multi-view priors and training characteristics of self-supervised learning, we propound an extremely efficient and stable training objective based on Gaussian mutual information maximization, exhibiting unprecedented efficiency compared to previous contrastive methods. Most significantly, extensive investigations in contexts beyond normal distributions demonstrate capability of our method to generalize to non-Gaussian scenarios.
- We bridge decorrelation-based self-supervised methods to our proposed contrastive objective from two aspects of approximation of form and consistency of solution, which points out a clue to demystify the relationships between various self-supervised learning methods.
- Thorough empirical studies demonstrate the effectiveness and efficiency of our method compared with advanced peers. Additionally, exploratory studies and visual analysis further reveal the advantages of our method and reinforce the understanding of our theoretical achievements.

## 2 RELATED WORK

### 2.1 Graph Self-Supervised Learning

For its remarkable performance, multi-view-based methods have been the dominant paradigm of graph self-supervised learning, which expect to explore common information from various augmented versions. A crucial aspect of these methods is to prevent degenerate solutions, where all representations are collapsed to a constant point (*i.e.*, complete collapse) or a subspace (*i.e.*, dimensional collapse) of the entire representation space. The current methods can be categorized into two groups, namely contrastive [20, 34, 55, 58, 59] and non-contrastive [6, 46, 57] approaches, based on their ways to circumvent model collapse.

The contrastive-based methods usually follow the criterion of mutual information maximization [21, 28], whose objective functions take the form of contrasting positive pairs with negative ones.

As pioneer works, DGI [49] and InfoGraph [45] learn unsupervised representations by maximizing mutual information between node-level representations and a whole graph summary vector based on the Jenson-Shannon estimator [31]. GraphCL [55], GRACE [58], and GCA [59] embed the *InfoNCE* [18] loss into graph contrastive learning framework. From the view of information theory, InfoGCL [54] investigates how to build appropriate contrastive learning frameworks for specific tasks. GII [53] treats the structure as a separate view and realizes cross-modal information interaction between features and structure. $\mathcal{M}$-ILBO [30] leverages MI estimators to maximize entropy for learning diverse representations.

The non-contrastive methods discard negative samples, which require special strategies to avoid collapsed solutions. BGRL [46] utilizes asymmetric architecture and a stop-gradient strategy to prevent the two branches from merging. Graph Barlow Twins (G-BT) [6] generalizes the celebrated Barlow Twins [56] from images to graph data. CCA-SSG [57] learns augmentation-invariant information while decorrelating features in different dimensions to prevent degenerated solutions.

### 2.2 Estimating Mutual Information

Mutual information is a powerful and commonly used measure for general correlation between random variables, which has been applied to a range of fields, including medical image processing [37], feature selection [1, 13], information bottleneck [16], and recommendation system [39]. Nevertheless, the exact computation of MI for high-dimensional variables is notoriously difficult. An alternative scheme is to estimate MI from empirical observations.

The non-parametric estimators make no assumptions about the underlying distribution of data and require no specification of any parameters. The most popular class in this branch is the k-nearest-neighbor-based estimators and their extensions [14, 26, 44]. Besides, the methods based on kernel density estimation (KDE) first estimate the probability density function and then compute MI by Monte-Carlo integration [40, 43].

The research on neural-network-based MI estimation [4, 18, 31] has also made significant process, which has been widely applied in representation learning. The key technical ingredient of these methods is to approximate the lower bound of MI based on dual representations of the $f$-divergence [31].

### 2.3 Collapse Issues in Self-Supervised Learning

Common types of collapse in multi-view self-supervised learning include complete collapse and dimensional collapse, which respectively represent the representations collapsing to a constant point and to a subspace. The most common approach to prevent collapse issues is to use negative samples to push data points apart in the representation space, including SimCLR [8], GRACE [58], and MVGRL [20]. Another class of methods employs asymmetric architectures with stop-gradient strategy to prevent representations of two views from colliding with each other [7, 9, 17, 47]. Besides, some methods aim to enhance representation diversity and prevent model collapse by evaluating and maximizing the entropy of representations. Representative works include CorInfoMax [32],

**Figure 1: An overview of the overall framework based on GMIM. The outputs of well-trained $f_\theta(\cdot)$ can be applied to various node-level downstream tasks. Best viewed in colors.**

$\mathcal{M}$-ILBO [30], and literature [50]. Furthermore, methods like VICReg [3] and CCA-SSG [57] decouple correlations between different representation channels to avoid dimensional collapse.

## 3 METHODOLOGY

### 3.1 Preliminaries and Overall Framework

*Preliminaries.* Before further discussion, the preliminary conceptions presented in this paper are first provided. A graph is denoted by $G(\mathbf{A}, \mathbf{X})$ with node set $\mathcal{V} = \{v_1, ..., v_N\}$ and edge set $\mathcal{E}$, where $|\mathcal{V}| = N$ indicates the number of nodes. Each node $v_i \in \mathcal{V}$ has a $D$-dimensional feature vector $\mathbf{x}_i \in \mathbb{R}^D$. Node feature matrix $\mathbf{X} = [\mathbf{x}_1, ..., \mathbf{x}_N]^\top \in \mathbb{R}^{N \times D}$ contains feature information of all nodes and adjacency matrix $\mathbf{A} \in \mathbb{R}^{N \times N}$ describes the connection relationship between different nodes. The task of node-level graph self-supervised learning is to seek good node representations $\widetilde{\mathbf{H}} = [\tilde{\mathbf{h}}_1, ..., \tilde{\mathbf{h}}_N]^\top \in \mathbb{R}^{N \times d}$ through learning a continuous mapping $f_\theta(\mathbf{A}, \mathbf{X}) : \mathbb{R}^{N \times N} \times \mathbb{R}^{N \times D} \to \mathbb{R}^{N \times d}$ without manual labels, where $\theta$ denotes learnable model parameters and $d$ indicates the representation dimension.

*Graph View Generation.* Let the transformation $\tau \in \mathcal{T} : G(\mathbf{A}, \mathbf{X}) \to G'(\mathbf{A}', \mathbf{X}')$ map the original graph to an augmented version, where $\mathcal{T}$ denotes the whole function space for augmentation. Specifically, the graph augmentation $\tau$ is jointly implemented from two aspects of graph topology and feature, following previous works [58]. For topology-level augmentation, *edge removal* is adopted, randomly removing edges of a certain ratio $p_e$ on the original graph. For feature-level augmentation, *node feature masking* randomly sets feature channels of a specific number $D \cdot p_f$ in feature matrix $\mathbf{X} \in \mathbb{R}^{N \times D}$ to zero, where $p_f$ is the masking ratio.

*Overall Framework.* In terms of basic framework, this paper inherits the common practice of prior studies. As shown in Figure 1, two various views $G'_A(\mathbf{A}'_A, \mathbf{X}'_A) = \tau_A(G)$ and $G'_B(\mathbf{A}'_B, \mathbf{X}'_B) = \tau_B(G)$ are generated based on two graph augmentation functions $\tau_A$ and $\tau_B$ randomly sampled from $\mathcal{T}$. The two augmented versions are

fed into a shared graph convolutional network [24] $f_\theta(\cdot)$ to obtain representations $\widetilde{\mathbf{H}}_A = [\tilde{\mathbf{h}}_1^A, ..., \tilde{\mathbf{h}}_N^A]^\top$ and $\widetilde{\mathbf{H}}_B = [\tilde{\mathbf{h}}_1^B, ..., \tilde{\mathbf{h}}_N^B]^\top$. To facilitate subsequent discussion, $\widetilde{\mathbf{H}}_A$ and $\widetilde{\mathbf{H}}_B$ are further batch-normalized into $\mathbf{H}_A = [\mathbf{h}_1^A, ..., \mathbf{h}_N^A]^\top$ and $\mathbf{H}_B = [\mathbf{h}_1^B, ..., \mathbf{h}_N^B]^\top$, each representation channel in which obey a distribution with 0-mean and 1-standard deviation. "GMIM" is the optimization objective proposed in the following sections.

### 3.2 Graph Self-supervised Learning with Gaussian Mutual Information Maximization

Contrastive learning is initially enlightened by the *InfoMax* principle [5], which expects to maximize mutual information between representations from various views.

**Definition 3.1** (Mutual Information). Let $X$ and $Y$ denote two $d$-dimensional continuous variables with marginal probability functions $p_x(X)$ and $p_y(Y)$, respectively. Their joint probability density is indicated by $p_{x,y}(X, Y)$. The mutual information $I(X; Y)$ between $X$ and $Y$ is defined as

$$I(X; Y) = \int_{\mathcal{X}} \int_{\mathcal{Y}} p_{x,y}(X, Y) \ln \frac{p_{x,y}(X, Y)}{p_x(X) \cdot p_y(Y)} dXdY, \quad (1)$$

where $\mathcal{X}$ and $\mathcal{Y}$ denote domains corresponding to $X$ and $Y$, respectively.

Nevertheless, the exact computation of mutual information for high-dimensional continuous variables is usually infeasible. First, it is challenging to estimate the probability densities from empirical observations. Second, even though they can be obtained, which may have complex forms, the integral operation in Eq. (1) remains difficult, even intractable. To tackle these issues, the conventional contrastive leaning methods employ parametric networks to directly estimate a lower bound of MI, which can be trained alongside the backbone via back-propagation in an end-to-end manner.

Divergent from the peer works, this paper assumes a latent Gaussian distribution for node representations and drops parametric estimators, which leads to a computationally tractable surrogate. The Gaussian assumption is justifiable and extensively employed in

numerous disciplines to simplify analysis and calculation, including economics, data science, and physics [33].

**Proposition 3.2** (Gaussian Mutual Information). *If the variables $X$ and $Y$ obey two multi-dimensional Gaussian distributions, respectively, the Gaussian mutual information $I_G(X;Y)$ between them is*

$$I_G(X;Y) = \frac{1}{2} \ln \frac{\det(\Sigma_X) \cdot \det(\Sigma_Y)}{\det(\Sigma_{X,Y})}, \qquad (2)$$

*where $\det(\cdot)$ indicates the determinant of a matrix, $\Sigma_X$ and $\Sigma_Y$ are the covariance matrices of $X$ and $Y$, respectively, and $\Sigma_{X,Y} = \begin{bmatrix} \Sigma_X & \Sigma_{XY} \\ \Sigma_{XY}^\top & \Sigma_Y \end{bmatrix}$ is the covariance matrix of variable $[X^\top, Y^\top]^\top$ with cross-covariance matrix $\Sigma_{XY}$.*

PROOF. Please refer to Section 1 of supplementary materials. □

The three covariance matrices $\Sigma_X$, $\Sigma_Y$, and $\Sigma_{X,Y}$ can be effortlessly estimated from the empirical data, which results in a straightforward calculation of Gaussian mutual information. The covariance matrix is a real symmetric matrix whose eigenvalues are all *greater than or equal to zero*. Mathematically, the determinant of a matrix is numerically equal to the product of all eigenvalues. Due to the underlying dimensional collapse issue during self-supervised pretraining, many eigenvalues of the empirical covariance matrix tend to be zero, which causes its determinant to approach zero. Therefore, a direct adoption of Eq. (2) for constructing a contrastive learning objective will bring about numerical instability. One feasible strategy to alleviate the numerical issue is to offset and scale the eigenvalues of the matrix performed by $\det(\cdot)$. Considering multi-view priors and training characteristics of SSL, a practical objective based on Gaussian Mutual Information Maximization (GMIM) can be formulated as

$$\mathcal{L}_{\text{GMIM}} = \ln \frac{\det(\mathbf{I} + \eta \cdot \Sigma_{A,B})}{\det(\mathbf{I} + \eta \cdot \Sigma_A) \cdot \det(\mathbf{I} + \eta \cdot \Sigma_B)}, \qquad (3)$$

where $\Sigma_A = \frac{1}{N} \mathbf{H}_A^\top \mathbf{H}_A, \Sigma_B = \frac{1}{N} \mathbf{H}_B^\top \mathbf{H}_B, \Sigma_{A,B} = \frac{1}{N} \begin{bmatrix} \mathbf{H}_A^\top \mathbf{H}_A & \mathbf{H}_A^\top \mathbf{H}_B \\ \mathbf{H}_B^\top \mathbf{H}_A & \mathbf{H}_B^\top \mathbf{H}_B \end{bmatrix}$, $\mathbf{I}$ is an identity matrix, and $\eta$ is a scaling factor with a typical value of 0.1. The eigenvalues of $\mathbf{I} + \eta \cdot \Sigma_A$ fall into $[1, +\infty)$, and so do the other two sibling matrices.

According to [10], the following property holds:

*Property* 1. For variables $X$ and $Y$, the relationship between entropy and mutual information is

$$I(X;Y) = H(X) - H(X|Y), \qquad (4)$$

where $H(X) = -\int_X p_x(X) \ln p_x(X) dX$ denote information entropy of $X$ under $p_x(X)$, and $H(X|Y) = \int_X \int_Y p_{x,y}(X,Y) \ln \frac{p_{x,y}(X,Y)}{p_y(Y)} dX dY$ is the conditional entropy of $X$ given $Y$. If $X$ is deterministic given $Y$, $H(X|Y) = 0$. Symmetrically, $I(X,Y) = H(Y) - H(Y|X)$ holds.

From Property 1, it can be known that mutual information maximization actually involves two potential processes: increasing information entropy and reducing conditional entropy. The conditional entropy is minimized when the relationship between $X$ and $Y$ can be described by a deterministic function $g(\cdot)$, that is, $Y' = g(X')$ holds for any pair $(X', Y') \sim p_{x,y}$. In our setup of overall framework, a shared graph neural network is employed, expecting that representations of different versions from the same instance can match

each other perfectly. In this circumstance, $g(\cdot)$ is preferred to be an identity mapping. By imposing the cross-view identity constraint to mutual information maximization with the preservation of entropy maximization, we can obtain an objective under Gaussian Mutual Information Maximization with Identity Constraint (GMIM-IC):

$$\mathcal{L}_{\text{GMIM-IC}} = \underbrace{\frac{1}{N} \sum_{v \in \mathcal{V}} \|\mathbf{h}_v^A - \mathbf{h}_v^B\|_2^2}_{\text{identity constraint}} - \underbrace{\sum_{* \in \{A,B\}} \beta \cdot \ln \det(\mathbf{I} + \eta \cdot \Sigma_*)}_{\text{entropy maximization}},$$
$$(5)$$

where $\beta$ is a coefficient balancing identity constraint term and entropy maximization term. Some analysis about Eq. (5) is placed in Section 2 of supplementary materials.

The objective functions $\mathcal{L}_{\text{GMIM-IC}}$ and $\mathcal{L}_{\text{GMIM}}$, which maximize Gaussian mutual information, can be computed directly in the representation space without relying on any additional architectures such as projection heads and estimators, demonstrating extremely high efficiency.

## 3.3 Gaussian Constraints

The proposed method is developed under the Gaussian assumption for node representations. The non-Gaussian nature of real-world scenarios may lead to misleading results in calculations and analyses conducted under Gaussian assumptions. Therefore, we design constraint functions from a maximum likelihood perspective to drive the actual distribution towards the target Gaussian distribution for alignment. The $j$-th column data in the representation matrix $\mathbf{H}_A$ can be viewed as $N$ empirical samples of a single-dimensional random variable. Let $\mu_A^j$ and $\sigma_A^j$ denote its mean and variance, respectively. A univariate Gaussian distribution $p_{gau}\left(x|\mu_A^j, \sigma_A^j\right)$ can be constructed, and by minimizing the negative log-likelihood $\sum_{i=1}^N -\log p_{gau}\left(H_{ij}^A|\mu_A^j, \sigma_A^j\right)$ where $H_{ij}^A$ denotes the element in the $i$-th row and $j$-th column of matrix $\mathbf{H}_A$, the $j$-th column data can be forced to approach a Gaussian distribution. Considering all representation channels from both views, the following objective function for Gaussian constraints can be constructed

$$\mathcal{L}_{gau} = \frac{1}{N \cdot d} \sum_{j=1}^d \sum_{i=1}^N -\log p_{gau}\left(H_A^{ij}|\mu_A^j, \sigma_A^j\right)$$
$$+ \frac{1}{N \cdot d} \sum_{j=1}^d \sum_{i=1}^N -\log p_{gau}\left(H_B^{ij}|\mu_B^j, \sigma_B^j\right). \qquad (6)$$

$\kappa \cdot \mathcal{L}_{gau}$ with a weighted coefficient $\kappa$ can serve as a probabilistic constraint loss, jointly supervising model training with $\mathcal{L}_{\text{GMIM}}$ or $\mathcal{L}_{\text{GMIM-IC}}$. Noting that subsequent empirical studies show that our method can achieve highly competitive results even without relying on $\mathcal{L}_{gau}$. However, $\mathcal{L}_{gau}$ remains worthwhile, which can act as a safeguard when our method fails in non-Gaussian scenarios.

## 4 BRIDGING CONTRASTIVE-BASED TO DECORRELATION-BASED

Based on the symbols in this article, the decorrelation-based self-supervised method (taking CCA-SSG [57] as an example) can be

formularized as

$$\mathcal{L}_{\text{CCA-SSG}} = \underbrace{\frac{1}{N}\|\mathbf{H}_A - \mathbf{H}_B\|_F^2}_{\text{invariance term}} + \lambda \cdot \underbrace{\left(\|\Sigma_A - \mathbf{I}\|_F^2 + \|\Sigma_B - \mathbf{I}\|_F^2\right)}_{\text{decorrelation term}}, \quad (7)$$

where $\lambda$ denotes a balancing factor and $\|\cdot\|_F$ indicates the Frobenius norm of a matrix. Since the diagonal elements of $\Sigma_A$ are always 1, the following equation holds:

$$\|\Sigma_A - \mathbf{I}\|_F^2 = \sum_{i=1}^{d}\sum_{j=1, j\neq i}^{d}(\Sigma_A^{ij})^2, \quad (8)$$

where $\Sigma_A^{ij}$ represents the element in the $i$-th row and the $j$-th column of $\Sigma_A$. The conclusion of Eq. (8) still holds for $B$. Next, we will establish connections between the decorrelation-based methods and our objective $\mathcal{L}_{\text{GMIM-IC}}$ from two perspectives.

### 4.1 Explaination 1: Approximate Form

**Lemma 4.1.** *For a square matrix* $\mathbf{M}$, $\det(\exp(\mathbf{M})) = \exp(\text{tr}(\mathbf{M}))$. *Replace* $\mathbf{M}$ *with* $\ln(\mathbf{I} + \eta \cdot \Sigma_*)$ :

$$\ln\det(\mathbf{I} + \eta \cdot \Sigma_*) = \text{tr}(\ln(\mathbf{I} + \eta \cdot \Sigma_*)), \quad (9)$$

*where* $* \in \{A, B\}$ [1]. *Applying Taylor expression to the logarithmic function in* $\text{tr}(\ln(\mathbf{I} + \eta \cdot \Sigma_*))$, *it can be known that*

$$\ln\det(\mathbf{I} + \eta \cdot \Sigma_*) = tr\left(\sum_{k=1}^{+\infty}\frac{(-1)^{k+1}}{k}(\eta \cdot \Sigma_*)^k\right). \quad (10)$$

Based on Lemma 4.1, we can obtain a second-order Taylor approximation:

$$-\ln\det(\mathbf{I} + \eta \cdot \Sigma_A) \approx \frac{\eta^2}{2} \cdot \sum_{i=1}^{d}\sum_{j=1, j\neq i}^{d}(\Sigma_A^{ij})^2 + \frac{\eta^2}{2} \cdot d - \eta \cdot d. \quad (11)$$

The proof of Lemma 4.1 and detailed derivations of Eq. (11) are placed in Section 3 of supplementary materials.

Comparing Eq. (8) with Eq. (11), $\|\Sigma_A - \mathbf{I}\|_F^2$ is equivalent to the second-order Taylor expression of $-\ln\det(\mathbf{I} + \eta \cdot \Sigma_A)$ without considering the constant term. Symmetrically, the finding can be extended to view $B$. Besides, the invariance term in Eq. (7) has an identical form with the identity constraint term in Eq. (5). Thus, we can conclude that the objective of decorrelation-based methods such as CCA-SSG has a approximate form with that of GMIM-IC.

### 4.2 Explaination 2: Consistent Solution

Certainly, the objective function in Eq. (7) is minimized when the representations from the two views are perfectly matched and their empirical covariance matrices tend towards the identity matrix.

**Proposition 4.2.** *When* $\ln\det(\mathbf{I}+\eta\cdot\Sigma_*)$ *or* $\ln\det(\Sigma_*)$ *is maximized, the empirical covariance matrix* $\Sigma_*$ *will converge to an identity matrix.*

PROOF. Refer to Section 3.2 of supplementary materials. □

Obviously, the identity constraint term is minimized in Eq. (5) when $\mathbf{H}_A$ and $\mathbf{H}_B$ is completely aligned. Combining this observation with Proposition 4.2, it can be concluded that the decorrelation-based objective in Eq. (7) has the same solution as the objective based on GMIM-IC.

---

[1] In the remaining sections of this article, $*$ is used to represent either $A$ or $B$.

Explaination 1 and 2 demonstrate the relationship between two objectives $\mathcal{L}_{\text{CCA-SSG}}$ and $\mathcal{L}_{\text{GMIM-IC}}$ from two aspects of approximation in form and consistency in final solutions. Consequently, the following remark emerges naturally.

*Remark* 4.3. The decorrelation-based graph self-supervised methods, which expect to align multiple views and disentangle different representation dimensions, can actually be viewed as a special instance of mutual-information-maximization-based contrastive learning under the Gaussian assumption and identity constraint.

## 5 THEORETICAL ANALYSIS

### 5.1 Preventing Dimensional Collapse

When dimensional collapse issue exists, various representation channels are coupled to each other and present a certain correlation. Another manifestation of dimensional collapse is that data points exhibit differences in distributions along different principal directions, where some directions exhibit loose distributions with higher variance, while others present tight distributions with lower variance.

*Property* 2. For empirical covariance matrix $\Sigma = \frac{1}{N}\mathbf{H}^\top\mathbf{H} \in \mathbb{R}^{d\times d}$ with batch-normalized representations $\mathbf{H} = [\mathbf{h}_1, ..., \mathbf{h}_N]^\top \in \mathbb{R}^{N\times d}$, which has $d$ eigenvalues $[\lambda_1, \lambda_2, \ldots, \lambda_d]$ corresponding to $d$ eigenvectors $[\mathbf{q}_1, \mathbf{q}_2, \ldots, \mathbf{q}_d]$, the variance of data $\mathbf{H}$ along the $k$-th principal direction (that is, direction of $\mathbf{q}_k$) is numerically equal to $\lambda_k$.

PROOF. Refer to Section 3.3 of supplementary materials. □

Property 2 potentially suggests that the unevenness of the eigenvalues of the covariance matrix leads to the issue of dimensional collapse. Combining with Proposition 4.2, it can be known that maximizing the logarithm of determinant can ensure entropy maximization and realize isotropic covariance, which actually guarantees the evenness of eigenvalues of the covariance matrix and thus prevents dimensional collapse issue. From the perspective of representation learning, this result will enhance the diversity, richness, and discriminability of node representations, thereby conferring advantages to downstream tasks.

### 5.2 Relation with *InfoNCE*

As the commonest indicator in contrastive learning, the *InfoNCE* [18] loss guides the model to learn meaningful and diverse representations by pulling together embeddings from positive pairs and pushing apart those from negative ones on the unit hypersphere.

A previous work [52] decomposes the classical *InfoNCE* objective into two terms: alignment term and uniformity term. The alignment term expects to match two views, which shares the same purpose as our identity constraint. The uniformity term is utilized to distribute representations uniformly on the unit hypersphere $\mathcal{S}^{d-1}$.

**Proposition 5.1.** *When the representations scatter over the unit hypersphere* $\mathcal{S}^{d-1}$ *uniformly (that is, they obey a complete uniform distribution), their entropy will reach the maximum value.*

PROOF. Refer to Section 3.4 of supplementary materials. □

Proposition 5.1 suggests that the uniformity term implicitly realize the maximization of entropy by distributing the representations uniformly over the hypersphere. Similar to the literature [52], some previous works [11, 50] also utilize the same entropy maximization

**Table 1: Node classification accuracy with standard deviation in percentage on six datasets. The "Input" column illustrates the data used in the training stage, and Y denotes labels. The bold font highlights the best results. "OOM" means Out-Of-Memory. For GII, the adjacency matrix is adopted as its structure view.**

| | Algorithm | Input | Cora | Citeseer | Pubmed | Computers | Photo | Coauthor-CS |
|---|---|---|---|---|---|---|---|---|
| | MLP | X, Y | 57.8 ± 0.2 | 54.2 ± 0.1 | 72.8 ± 0.2 | 79.81 ± 0.06 | 86.36 ± 0.08 | 91.32 ± 0.11 |
| | GCN | X, A, Y | 81.5 | 70.3 | 79.0 | 86.51 ± 0.54 | 92.42 ± 0.22 | 93.03 ± 0.31 |
| | GAT | X, A, Y | 83.0 ± 0.7 | 72.5 ± 0.7 | 79.0 ± 0.3 | 86.93 ± 0.29 | 92.56 ± 0.35 | 92.31 ± 0.24 |
| Unsupervised | DeepWalk | A | 68.5 ± 0.5 | 49.8 ± 0.2 | 66.2 ± 0.7 | 85.68 ± 0.06 | 89.44 ± 0.11 | 84.61 ± 0.22 |
| | GAE | X, A | 72.1 ± 0.5 | 66.5 ± 0.4 | 71.8 ± 0.6 | 85.27 ± 0.19 | 91.62 ± 0.13 | 90.01 ± 0.71 |
| | GMI | X, A | 83.0 ± 0.3 | 72.4 ± 0.1 | 79.9 ± 0.2 | 82.21 ± 0.31 | 90.68 ± 0.17 | OOM |
| | GRACE | X, A | 81.9 ± 0.4 | 71.3 ± 0.3 | 80.1 ± 0.2 | 86.53 ± 0.28 | 92.24 ± 0.17 | 92.98 ± 0.05 |
| | GCA | X, A | 81.7 ± 0.3 | 71.1 ± 0.4 | 79.5 ± 0.5 | 87.85 ± 0.31 | 92.49 ± 0.09 | 93.10 ± 0.01 |
| | GraphMAE | X, A | 84.2 ± 0.4 | 73.4 ± 0.4 | 81.1 ± 0.4 | 88.12 ± 0.30 | 92.97 ± 0.21 | 93.03 ± 0.16 |
| | G-BT | X, A | 84.0 ± 0.4 | 73.0 ± 0.3 | 80.7 ± 0.4 | 88.14 ± 0.33 | 92.63 ± 0.44 | 92.95 ± 0.17 |
| | CCA-SSG | X, A | 84.2 ± 0.4 | 73.1 ± 0.3 | 81.6 ± 0.4 | 88.74 ± 0.28 | 93.14 ± 0.14 | 93.31 ± 0.22 |
| | InfoGCL | X, A | 83.5 ± 0.3 | 73.5 ± 0.4 | 79.1 ± 0.2 | - | - | - |
| | $GII_{l-g}$ | X, A | 83.5 ± 0.6 | 73.2 ± 0.4 | 79.5 ± 0.3 | - | - | - |
| | CorInfoMax | X, A | 82.6 ± 0.4 | 72.2 ± 0.5 | 80.4 ± 0.4 | 87.98 ± 0.14 | 92.63 ± 0.10 | 92.88 ± 0.15 |
| | $\mathcal{M}$-ILBO | X, A | 84.3 ± 0.5 | 73.2 ± 0.7 | 81.4 ± 0.5 | 88.76 ± 0.31 | 93.06 ± 0.31 | 93.14 ± 0.26 |
| | MVGRL | X, A | 83.7 ± 0.6 | **73.6 ± 0.3** | 79.9 ± 0.2 | 87.52 ± 0.11 | 91.74 ± 0.07 | 92.11 ± 0.12 |
| | DGI | X, A | 82.3 ± 0.6 | 71.8 ± 0.7 | 76.8 ± 0.6 | 83.95 ± 0.47 | 91.61 ± 0.22 | 92.15 ± 0.63 |
| | GMIM | X, A | 83.3 ± 0.5 | 72.6 ± 0.6 | 81.0 ± 0.7 | 88.71 ± 0.36 | 92.84 ± 0.16 | 92.67 ± 0.11 |
| | GMIM-IC | X, A | **84.5 ± 0.5** | **73.6 ± 0.4** | **81.8 ± 0.6** | **89.04 ± 0.35** | **93.17 ± 0.27** | **93.47 ± 0.23** |

criterion to promote uniformity and diversity of representations. In contrast, our method explicitly maximizes the entropy of representations under the assumption of Gaussian distribution. In general, the two approaches reach the similar goal by different routes.

## 6 EXPERIMENTS

### 6.1 Datasets and Experimental Setup

*Datasets.* To assess our approach, six widely used benchmark datasets are adopted for experimental study, including three citation networks **Cora**, **Citeseer**, and **Pubmed** [41], two co-purchase networks **Amazon-Computers** and **Amazon-Photo** [42], and one co-authorship network **Coauthor-CS** [42].

*Experimental Setup.* The representation encoder is implemented by Graph Convolutional Network (GCN) [24]. The model parameters are initialized via Xavier initialization [15] and trained by Adam optimizer [23]. All experiments are conducted on a NVIDIA RTX 3090 GPU with 24 GB memory. The representations are first learned through our method in an unsupervised way and then evaluated by a simple linear classifier, which is the most common manner in the current self-supervised learning literature.

### 6.2 Comparison Experiments

Here, we compare our method with state-of-the-art baselines in terms of performance and efficiency.

*Performance Comparison.* To evaluate the effectiveness of our approach, we compare our method with the state-of-the-art baselines on node classification task under the simple linear classifier. The average classification accuracy with standard deviation of 20

results is reported for each dataset. We compare our approach with unsupervised methods including DeepWalk [35], GAE [25], DGI [49], GMI [34], GRACE [58], GCA [59], G-BT [6], CCA-SSG [57] InfoGCL [54], GII [53], GraphMAE [22], CorInfoMax [32], $\mathcal{M}$-ILBO [30], and MVGRL [20]. Furthermore, some supervised models including multi-layer perceptron (MLP), GCN [24], and GAT [48] are also as baselines. We adopt the public splits on Cora, Citeseer and Pubmed, and a 1:1:8 split for training/validation/testing on the other three datasets. To make a fair comparison, for the methods without adopting the same splits as ours, we conduct experiments to get relevant results based on the officially released source code with a hyper-parameter search. Table 1 reports the classification results on six datasets. It can be observed that our method achieves high performance on all datasets and outperforms the state-of-the-art peers. In particular, our method significantly outperforms neural estimator-based methods such as GRACE, GCA, and InfoGCL. These results clearly demonstrate the effectiveness of our approach. After subjecting the node representations to a rigorous statistical hypothesis testing, we discover that they do not actually conform to a Gaussian distribution. In other words, our method remains highly effective in non-Gaussian scenarios. Overall, GMIM-IC surpasses GMIM. One reason is that the identity constraint imposes stricter demands on cross-view consistency, which aligns with the practical design of the shared network architecture. Besides, GMIM-IC demonstrates comparable performance with CCA-SSG, which can serve as empirical support for our theoretical analysis.

*Efficiency Comparison.* To illustrate the simplicity and efficiency of our model, we compare our method with other graph contrastive methods based on mutual information estimators in terms of numbers of model parameters, time consumption of training stage, and

**Table 2: Comparison of numbers of model parameters, training time, and memory costs between various graph contrastive methods. The term "Paras" denotes the number of model parameters. For MVGRL, the representation dimensions on Pubmed and Amazon-Computers are set to 256 and 512, respectively. For DGI, the representation dimension on Amazon-Computers is set to 512. For GMIM and GMIM-IC, the output dimensions are set to 512 across the four datasets.**

| Algorithm | Cora | | | Citeseer | | | Pubmed | | | Computers | | |
|---|---|---|---|---|---|---|---|---|---|---|---|---|
| | Paras | Time | Memory | Paras | Time | Memory | Paras | Time | Memory | Paras | Time | Memory |
| DGI | 996K | 6.8s | 3.8GB | 2158K | 9.4s | 7.8GB | 194K | 44.9s | 11.2GB | 1,808K | 71.2s | 11.3GB |
| GRACE | 433K | 5.1s | 1.2GB | 2,159K | 7.4s | 1.5GB | 519K | 1,169s | 12.2GB | 263K | 362.8s | 7.4GB |
| MVGRL | 1,731K | 23.7s | 3.8GB | 4,055K | 48.4s | 7.9GB | 322K | 2,010s | 9.1GB | 1,049K | 78.8s | 16.6GB |
| GMIM | 997K | 2.8s | 2.5GB | 1,896K | 2.5s | 2.6GB | 519K | 9.5s | 3.4GB | 656K | 7.5s | 3.2GB |
| GMIM-IC | 997K | 3.1s | 2.5GB | 1,896K | 2.9s | 2.6GB | 519K | 7.2s | 3.4GB | 656K | 8.7s | 3.2GB |

---

**Algorithm 1** Hypothesis Testing based on scipy.

```
import numpy as np
from scipy import stats
# H: node representation matrix with the size of (N, d)
ret = stats.normaltest(H, axis=0)[1] # results the shape of (d,)
# The p-value of hypothesis testing on Four datasets are:
# Cora: [2.34e-46, 3.01e-84, ..., 1.01e-52, 1.63e-47, 3.18e-40]
# Citeseer: [1.29e-12, 1.46e-08, ..., 3.22e-05, 4.22e-03, 3.72e-07]
# Pubmed: [9.06e-34, 3.89e-118, ..., 5.18e-131, 3.09e-48, 8.23e-72]
# Computers: [1.07e-19, 1.97e-23, ..., 3.81e-07, 1.36e-08, 4.82e-36]
```

memory costs. Table 2 summarizes all indicators of various methods. Overall, compared to other methods, our method has fewer model parameters, shorter training time, and smaller memory costs in most cases. This is because our method doesn't rely on additional projection heads, parameterized mutual information estimator, and negative samples, which add extra calculation, additional parameters, and storage burden. Besides, the short training time potentially indicates the fast convergence of our algorithm. The simplicity of our model and the efficiency of the calculation of objective function significantly reduce the time and space complexity of our method.

## 6.3 Gaussian Testing and Effect of Gaussian Constraints

*Histograms and Hypothesis Testing for Node Representations.* The histograms of node representations are illustrated in Figure 2. At first glance, the distribution of representations exhibits a Gaussian appearance. This observation served as the initial motivation of our research and sparked our curiosity about the possibility of directly performing graph self-supervised learning under Gaussian mutual information maximization. In general circumstances, mutual information cannot be directly computed and the current contrastive learning methods rely on additional neural estimators to approximate a lower bound. Without disappointment, empirical results in Table 1 demonstrate the effectiveness of our approach. Subsequently, we conduct a rigorous hypothesis testing on individual channels of representation matrices of multiple datasets based on library scipy, as shown in Algorithm 1. The outcomes indicate that node representations do not actually conform to a Gaussian distribution. This result is, in fact, promising, which implies that our approach will no longer be confined to Gaussian scenarios. In summary, visualized histograms and the Gaussian assumption provided the initial impetus for our research, while the fact that

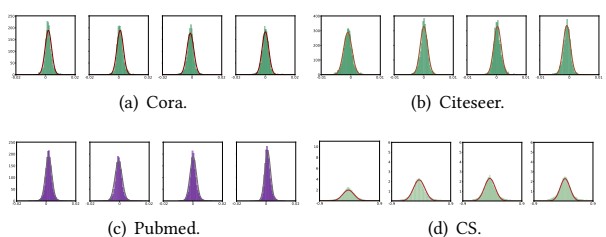

(a) Cora.          (b) Citeseer.

(c) Pubmed.          (d) CS.

**Figure 2: Histograms of individual representation channels on four datasets. The curve in each subfigure represents a Gaussian distribution with mean and variance from the corresponding histogram. The histograms appear to exhibit a Gaussian appearance.**

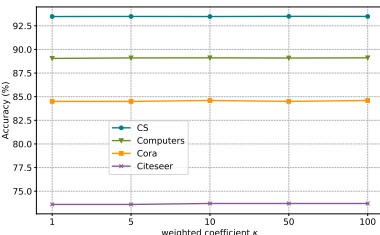

**Figure 3: Effect of Gaussian constraints under GMIM-IC.**

our approach still remains its performance under the non-Gaussian conditions extends the application scenarios of our method.

*Effect of Gaussian Constraints.* We test the effect of the Gaussian constraints $\mathcal{L}_{gau}$ in Eq. (6) on four datasets: Cora, Citeseer, Computers, and CS, which is attached to $\mathcal{L}_{\text{GMIM-IC}}$ with a weighted coefficient $\kappa$. As shown in Figure 3, we can find that $\mathcal{L}_{gau}$ hardly improves the performance of our method on these existing datasets. This is because our method has already achieved highly competitive results on these datasets, unaffected by non-normality.

## 6.4 Hyperparameter Sensitivity Analysis and Exploratory Experiments

*Effect of Representation Dimension.* We conduct experiments by varying the representation dimension to investigate its impacts on performance. Figure 4 summarizes the results of the three variants

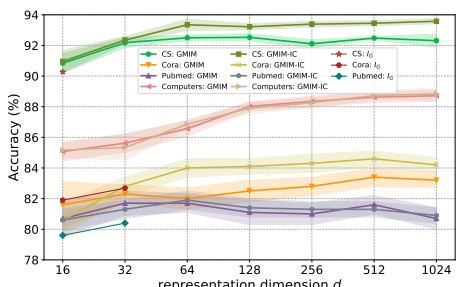

Figure 4: Effect of representation dimension. "$I_G$" denotes the results based on Eq. (2).

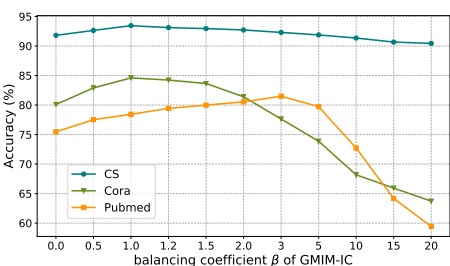

Figure 5: The classification accuracy of GMIM-IC under varying balancing coefficients.

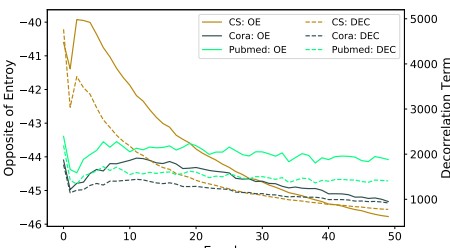

Figure 6: Synergistic changes between opposite of entropy (OE) and decorrelation loss (DEC).

based on Eq. (2), Eq. (3), and Eq. (5) on four datasets. It can be observed that our method achieves optimal performance with an appropriately large dimension, because the representations exhibit better discriminability and linear separability in high-dimensional space. However, as the dimension becomes excessively large such as 1,024, there is a slight decrease in performance. This can be blamed on the fact that an excessively high representation dimension hinders the model from learning compact and information-dense representations. Another non-negligible underlying factor for declining performance is that higher dimensions lead to poorer estimation of the covariance matrix. Even in low-dimensional settings, our method still delivers decent performance. This finding can be attributed to the effective maximization of information entropy, which prevents dimensional collapse, enhances the diversity of representations, and ultimately improve model performance within limited dimensions. Under the objective function based on Eq. (2), the results in high-dimensional settings and on Computers are unavailable. In such scenarios, the covariance matrix exhibits numerous small eigenvalues, causing its determinant to approach zero. This fact introduces numerical instability and eventually disrupts training process.

*Impact of Balancing Coefficient.* We study the impacts of the balancing coefficient $\beta$ in $\mathcal{L}_{\text{GMIM-IC}}$ on performance. Figure 5 illustrates the variation of classification accuracy with varying values of the coefficient. The performance exhibits a pattern of initially increasing and later decreasing as $\beta$ goes up. When $\beta$ is small, the entropy maximization term cannot fully exploit its role in promoting diversity of representations. When $\beta$ is too large, too much emphasis on maximizing information entropy leads to informative yet meaningless representations.

*Synergistic Changes Between Opposite of Entropy and Decorrelation Loss.* Taking $\mathcal{L}_{\text{GMIM-IC}}$ as the optimization objective, we visualize the joint changes of decorrelation loss in Eq. (7) and opposite of entropy in Eq. (5). For each dataset in Figure 6, the decorrelation loss (dashed line) exhibits a nearly identical trend to the opposite of entropy (solid line). Experimental observations potentially indicate a similar effect between them, which can serve as an empirical support for Section 4.

**More experimental results and visual analysis are provided in the supplementary material.**

## 7 LIMITATIONS, CONCLUSION, AND FUTURE WORK

*Limitations.* Due to extreme limitations in computational resources, we only conducted empirical studies on graphs. Extension experiments on other types of data, such as images or multimodal data, are left for future, which relies on many GPUs.

*Conclusion.* In this paper, we have presented a graph contrastive learning method under the common Gaussian assumption for node representations, which does not rely on any parametric mutual information estimators and negative samples. Furthermore, we provide two theoretical explanations regarding the relationship between decorrelation-based methods and contrastive-based methods. Our analysis reveals that the decorrelation-based method can be interpreted as a variant of contrastive methods when the Gaussian assumption and identity constraint are considered. Extensive comparative experiments and visual analysis have demonstrated the effectiveness, efficiency, and theoretical soundness of our method. Overall, the Gaussian assumption motivates our research, but empirical evidence demonstrates the continued effectiveness of our method in non-Gaussian scenarios, which significantly extends the practical application scope of our work.

*Future Work.* Our research paves a new path for graph self-supervised learning. The prospect of extending the Gaussian assumption to other distributions, such as the Cauchy distribution, stands as a viable endeavor. Furthermore, the exploration of relationships among distinct variants under different distributions represents a valuable and exciting pursuit.

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
