# OpenReview forum: "Gaussian Mutual Information Maximization for Efficient Graph Self-Supervised Learning: Bridging Contrastive-based to Decorrelation-based"
_acmmm.org/ACMMM/2024/Conference — MM2024 Poster_

### Official Review · Reviewer_g4gt · 2024-05-21

**Rating:** 5
**Confidence:** 4

**Summary:**

The paper proposes a novel graph contrastive learning method, introducing an innovative approach in the field. The motivation behind the method is well-founded, and the approach itself is both creative and effective. The theoretical analysis conducted by the authors is accurate, although deriving it may not be particularly challenging.

**Strengths:**

- The article explores the information maximization approach for graph self-supervised learning effectively.
- The paper introduces a novel graph contrastive learning algorithm and offers comprehensive numerical experiments for graph self-supervised learning.
- The article explores the information maximization approach for graph self-supervised learning.
- It offers comprehensive numerical experiments for graph self-supervised learning.
- The theoretical analysis is well done.
- A novel graph contrastive learning algorithm is introduced.
- The proposed method achieves comparable results to existing approaches.

**Limitations:**

- It is reasonable to assume that the error between $h^A$ and $h^B$  follows a Gaussian distribution, but is each of them also Gaussian? In most contrastive losses, the second term (or the negative term) is typically an expectation, a mean, or a norm. However, the main concept that makes sense is the first term, which can be easily modeled as Mean Squared Error (MSE).
- The code implementation shows significant similarities to CCA-SSG.
- Title of the paper is too long.

**Suitability:**

3

---

### Official Review · Reviewer_DneH · 2024-05-24

**Rating:** 3
**Confidence:** 4

**Summary:**

The paper proposed a GCL method that eliminates the reliance on parameterized estimators and negative samples based on the maximization of Gaussian mutual information. The proposed method tries to bridge the contrastive-based self-supervised method to the decorrelation-based self-supervised method, and comes up with a relatively simple framework. Experiments are done to measure the performance of the method.

**Strengths:**

- The paper is well-written and easy to follow.
- The proposed method does not rely on structures like projection head, parameterized estimators, and negative samples, while still showing good performance.
- The theoretical part of the method is derivated rigorously.

**Limitations:**

There are some problems with the experimental part:
As for the performance comparison part:
- Experiments are only done on node classification.
- Accuracy on Computers, Photo, and Coauthor-CS is displayed with two decimals, while accuracy on the rest three datasets is not.
- While method GMI on Coauthor-CS is marked 'OOM', InfroGCL and GII on Computers, Photo, and Coauthor-CS is marked '-' without explanation
- There are some data given without standard
As for the efficiency comparison part:
The method shows fewer costs in various respects. However, the experiments are somewhat incomplete.
- The comparison is done with 'other graph contrastive methods based on mutual information estimators', while not all methods of this kind are compared. For example, GMI, which is the basis of the proposed method, and M-ILBO, which shows relatively good performance and is discussed in section 6.3 of supplementary, are not tested in the efficiency comparison part.
- As the paper claims to bridge the gap between two research areas, the comparison may not be restricted by the kind of method.

Furthermore, there are some minor questions:
- The theory is based on the Gaussian condition while still achieving good results on the non-Gaussian condition. Is there any explanation?
- Or will $L_{gau}$ decrease even without adding it to loss?

**Suitability:**

2

---

### Official Review · Reviewer_BFyx · 2024-05-24

**Rating:** 3
**Confidence:** 2

**Summary:**

The paper outlines the advancements in Graph Contrastive Learning , particularly focusing on the introduction of Gaussian Mutual Information (GMI) as a computationally efficient alternative to conventional methods. It highlights the bridging of GMI with decorrelation-based self-supervised learning, proposing a unified framework. Empirical evidence highlighting the superiority of the method over other methods.

**Strengths:**

- The paper is well-written and easy to follow. Self-supervised learning is still a hot topic in the current community.
- I noticed that the comments raised by other reviewers during the previous conference review process were well addressed. In the appendix, the paper provides detailed description about proof, experimental setting and necessary hyperparameter configuration.

**Limitations:**

- It seems the topic is out of scope of multimedia. Although this is a relatively complete work, I would recommend submitting it to other suitable machine learning conferences.
- The term 'efficient' in the title doesn't appear to be formally addressed in the paper. I observed that Table 2 includes a time comparison with other methods. This is good, but why not include the time cost for some methods, e.g. CCA-SSG, mentioned in Table 1 and compare them? This will make the experimental results more convincing.

Minor points:
- Some of the statements in the paper are a bit long-winded, such as line 369 "Mathematically, the determinant of a matrix is ​​numerically equal to the product of all eigenvalues", and readers should have this basic knowledge. Another similar situation is lines 398-400. t is recommended that the author simplify the writing appropriately.

**Suitability:**

1

---

### Official Review · Reviewer_6uFX · 2024-05-27

**Rating:** 4
**Confidence:** 4

**Summary:**

The paper introduces a method for graph self-supervised learning (SSL), focusing on Gaussian Mutual Information Maximization (GMIM) to enhance the efficiency and effectiveness of learning representations from graph data. The authors develop a theoretically motivated approach that simplifies mutual information estimation by assuming Gaussian distributions for node representations, thereby avoiding the complexities of parameterized mutual information estimators and the need for negative samples. This method also bridges the gap between contrastive and decorrelation-based SSL methods by demonstrating their theoretical and practical similarities under Gaussian assumptions.

**Strengths:**

*  The introduction of GMIM for graph SSL is interesting;
* The authors provide a rigorous explanation of how the method relates to and unifies different branches of SSL methodologies;
* The method is thoroughly evaluated against several benchmarks and state-of-the-art approaches on multiple datasets.

**Limitations:**

* While the Gaussian assumption simplifies the computation, it may not always hold in real-world data, which could limit the applicability of the method in scenarios where the data distribution significantly deviates from Gaussian;
* The performance of the method appears sensitive to the choice of representation dimension and balancing coefficients, which may require careful tuning in different applications;
*  The paper does not discuss the long-term computational costs associated with scaling to very large graphs, which could be a concern for practical implementations in large-scale applications.

**Suitability:**

2

---

### Meta-Review · Area_Chair_XdTy · 2024-07-02

**Recommendation:** Accept (Poster)
**Confidence:** 3

**Metareview:**

The reviewers agree that this is a well-written submission. 6uFX notes that the motivation for the method is rigorous and DneH and g4gt echo this comment. BFyx notes that self-supervised learning is a hot topic. G4gt, Dneh, and 6uFX suggest that the evaluation demonstrates good performance.

6uFX and G4Gt critique the Gaussian assumption for data. 6uFx also suggests that there could be issues with tuning parameters and computational costs on large graphs. This suggestion is picked up by BFyx who notes that the work does not really centre the term “efficient” that appears in the title. DneH has a long list of limitations, some of which are minor but calls the focus on node classification a limitation here.

The ratings here are overall on the positive side of borderline with a range from weak accept to borderline reject. Although the suitability of the work ranges in the review from “not suitable” to “definitely suitable”, the authors admit that multimodal application would be future work and relies on being “easily extended”. (Well, if it’s so easy, why didn’t the authors do it?)

Given the renewed emphasis on distinguishing this venue from others by centring clearly multimedia/multimodal research rather than research which *may* be applied to multimedia data, this limitation must be serious and hold back enthusiasm for this work.